# Curcumin Derivative C66 Suppresses Pancreatic Cancer Progression through the Inhibition of JNK-Mediated Inflammation

**DOI:** 10.3390/molecules27103076

**Published:** 2022-05-11

**Authors:** Hongjin Chen, Yuchen Jiang, Rongdiao Liu, Jie Deng, Qinbo Chen, Lingfeng Chen, Guang Liang, Xiong Chen, Zheng Xu

**Affiliations:** 1Translational Medicine Research Center, Guizhou Medical University, Guiyang 550000, China; chicaihecha@163.com; 2The Fourth Affiliated Hospital, Zhejiang University School of Medicine, Yiwu 322000, China; jycyx2022@163.com; 3Clinical Pharmacy Center, Department of Pharmacy, Zhejiang Provincial People’s Hospital, Affiliated People’s Hospital Hangzhou Medical College, Hangzhou 310000, China; 4Key Laboratory of Diagnosis and Treatment of Severe Hepato-Pancreatic Diseases of Zhejiang Province, The First Affiliated Hospital, Wenzhou Medical University, Wenzhou 325000, China; rliu@wmu.edu.cn (R.L.); dengjiewz@163.com (J.D.); 18767755785@163.com (Q.C.); 5School of Pharmacy, Hangzhou Medical College, Hangzhou 310000, China; lfchen@hmc.edu.cn (L.C.); wzmcliangguang@163.com (G.L.); 6Department of Endocrinology, The First Affiliated Hospital, Wenzhou Medical University, Wenzhou 325000, China

**Keywords:** C66, pancreatic cancer, inflammation, JNK

## Abstract

Pancreatic adenocarcinoma is by far the deadliest type of cancer. Inflammation is one of the important risk factors in tumor development. However, it is not yet clear whether deterioration in pancreatic cancer patients is related to inflammation, as well as the underlying mechanism. In addition, JNK is abnormally activated in pancreatic cancer cells and the JNK inhibitor C66 reduces the inflammatory microenvironment in the tumor. Therefore, the aim of this study was to evaluate the role of C66 in the proliferation and migration of pancreatic cancer. Our results showed that various inflammatory cytokines, such as IL-1β, IL-6, IL-8, and IL-15, were more expressed in pancreatic cancer than in the matching normal tissue. Furthermore, C66, a curcumin analogue with good anti-inflammatory activity, inhibited the proliferation and migration of pancreatic cancer cells in a dose-dependent manner, and effectively inhibited the expression of the above inflammatory factors. Our previous research demonstrated that C66 prevents the inflammatory response by targeting JNK. Therefore, in this study, JNK activity in pancreatic cancer cells was investigated, revealing that JNK was highly activated, and the treatment with C66 inhibited the phosphorylation of JNK. Next, shJNK was used to knockdown JNK expression in pancreatic cancer cells to further confirm the role of JNK in the proliferation and migration of this tumor, as well as in the inflammatory tumor microenvironment (TME). The results demonstrated that JNK knockdown could significantly inhibit the proliferation and migration of pancreatic cancer. Moreover, the low JNK expression in pancreatic cancer cells significantly inhibited the expression of various inflammatory factors. These results indicated that C66 inhibited the progression of pancreatic cancer through the inhibition of JNK-mediated inflammation.

## 1. Introduction

Pancreatic adenocarcinoma (PAAD) is the most deadly malignant tumor of the digestive tract [1]. The incidence of PAAD increases year by year, and the 5-year survival rate is less than 5%. It is estimated that PAAD will be the second leading cause of cancer-related deaths by 2030 [2]. Unfortunately, routine examinations cannot be performed due to the inaccessibility of the anatomical location of the pancreas, and more than half of PAAD patients are diagnosed at an advanced stage, and are thus not suitable for surgical resection. In addition, although chemotherapy is the main treatment used to shrink or reduce the growth of this cancer, allowing patients to live longer, it is not curing it. Drug resistance is also a huge obstacle in the improvement of the overall survival, and nearly 80% of PAAD patients undergoing surgical resection are at risk of recurrence [1]. However, 80% of PAAD patients eventually die from tumor metastasis regardless of whether they have undergone surgical resection or not. At present, many studies have been conducted to cure PAAD, but no effective treatment strategy has been successful. Therefore, the study of the molecular mechanism of PAAD is essential to find new targets to cure it.

In recent years, many studies have reported that tumors and inflammation are closely related, and the inflammatory microenvironment can accelerate the occurrence and development of tumors [3]. An article reported that NF-κB, a classic inflammatory signaling pathway in PAAD, is highly activated, and the inhibition of this pathway can alleviate the proliferation and migration of PAAD cells [4]. TLR2 is one inflammatory biomarker that promotes tumor growth and is associated with patient survival and chemotherapy response in PAAD [5]. One meta-analysis indicated that inflammation is a vital index in pancreatic carcinoma patients [6]. However, it is not yet clear whether the deterioration of pancreatic cancer patients is related to inflammation, as well as the underlying mechanism.

Previous reports found that JNK, a member of the MAPK family, is in an abnormally activated state in PAAD [7]. The inhibition of JNK activity could inhibit the proliferation and migration of PAAD [8]. Nevertheless, it is still unknown whether JNK inhibits the inflammatory microenvironment of the tumors. Therefore, in this work, various inflammatory factors in PAAD were evaluated, revealing that IL-6, IL-1β, IL-8, and IL-15 were highly expressed in PAAD. Moreover, JNK was also abnormally activated in PAAD cells. Using the JNK inhibitor C66 reported in our previous study [9], a significant inhibition in the activity of JNK was found in PAAD cells. In addition, C66 reduced the inflammation in the tumor microenvironment (TME). JNK knockdown also exerted a significant inhibitory effect on the epithelial–mesenchymal transition in PAAD. Therefore, the aim of this study was to investigate the role of C66 in the inhibition of pancreatic cancer tissue inflammation and consequent proliferation and migration of PAAD by targeting JNK.

## 2. Results

### 2.1. High Expression of Inflammatory Factors in Pancreatic Cancer

The expression of IL-1β, IL-6, IL-8, and IL-15 was higher in PAAD than in normal tissues, as shown in Figure 1A. To investigate whether IL-6, the inflammatory marker, is highly expressed in PAAD, two cases (grade 2) of human PDAC tumor tissues and corresponding adjacent nontumor tissues were obtained from patients undergoing surgical resection of pancreatic cancer. The immunohistochemistry results in tumor samples of PAAD patients showed that IL-6 expression was significantly higher than that in the matching normal tissue (Figure 1B). The association between IL-6 expression and the overall survival of patients with PAAD was further evaluated to determine whether if its high level could play a role in tumor development. The survival curves indicated that PAAD patients with high IL-6 expression had a shorter overall survival. In contrast, patients with PAAD and low IL-6 expression had a longer survival (Figure 1C). Moreover, the three pancreatic cancer cell lines SW1990, PANC-1, and BxPC-3 showed a higher expression of IL-6 compared with its expression in the pancreatic normal cell line HPNE (Figure 1D). These results indicated that inflammation might participate in the deterioration of PAAD patients.

### 2.2. Inhibition of Inflammatory Factors by C66

Our previous study showed that C66 exerts an excellent anti-inflammatory effect (structure shown in Figure 2A). Additionally, the above result showed that IL-6 is highly expressed in PANC-1 and SW1990, compared with its expression in HPNE (2.0-fold and 3.2-fold, respectively). Thus, the effect of C66 in the release of inflammatory cytokines in PANC-1 and SW1990 cell lines was evaluated. The results showed that C66 decreased IL-6 expression both in PANC-1 and SW1990 in a dose-dependent manner (Figure 2B). Moreover, C66 concentrations attenuated IL-6 release (Figure 2C). Furthermore, the inflammatory factors IL-1β, IL-6, IL-8, IL-12β, IL-15, TNF-α, and COX2 were significantly decreased after treatment with C66 (Figure 2D,E).

### 2.3. Inhibition of the Proliferation and Migration of Pancreatic Cancer Cells by C66

PANC-1 and SW1990 growth was significantly inhibited after the treatment with C66 for 72 h (Figure 3A,B), with a IC50 of 113.4 μM (PANC-1) and 91.83 μM (SW1990). Furthermore, the colony assay demonstrated that, the higher the concentration of C66, the more potent the inhibition of cell proliferation (Figure 3C,D).

Because pancreatic cancer patients suffer from high mortality from metastasis, the migration ability of pancreatic cancer cells was also investigated by the scratch wound assay and the transwell migration assay in vitro, and the results demonstrated that C66 significantly reduced the ability of PANC-1 and SW1990 cells to migrate in a dose-dependent manner (Figure 4).

### 2.4. Reduction in Pancreatic Cancer Cell Inflammation by C66 Targeting JNK

Our previous study demonstrated that C66 targets JNK, playing a role as an anti-inflammatory compound. The present results showed that JNK activation in pancreatic cancer cell lines was increased as compared with the control cell line, HPNE (Figure 5A), and C66 markedly inhibited JNK phosphorylation in PANC-1 and SW1990 in a dose-dependent manner (Figure 5B,C).

Next, stable JNK knockdown in PANC-1 and SW1990 cell line was established by lentiviral transfection to clarify the role of JNK in the development of pancreatic cancer (Figure 6A,B). Further experiments performed to confirm JNK mediated inflammation in the TME demonstrated that IL-6 expression was remarkably decreased after JNK knockdown (Figure 6C). Additionally, IL-1β, IL-6, IL-8, IL-12β, IL-15, TNF-α, and COX2 mRNA expression was also suppressed by JNK silencing (Figure 6D). JNK knockdown could also inhibit pancreatic cancer cell proliferation and migration (Figure 6E–G). These results suggested that JNK might be one of the critical regulators of inflammatory TME.

## 3. Discussion

Pancreatic cancer is currently the tumor with the highest fatality rate worldwide. In recent years, many studies have been performed on the pathogenesis of pancreatic cancer, but its pathogenesis and key targets are still controversial and complicated. Inflammatory cancer transformation is a novel theory of tumor pathogenesis developed in recent years. Inflammatory TME has a direct causal relationship in a variety of tumors. NF-kB signaling pathway in pancreatic cancer is highly activated, and it can transcribe a variety of inflammatory factors to exacerbate tumor deterioration [4]. In addition, MAPK signaling pathway, another inflammatory signaling pathway, and its key genes p38 and JNK, are highly expressed in pancreatic cancer clinical samples and cells [10]. This study revealed that IL-6, IL-1beta, and other inflammatory factors are highly expressed in samples from pancreatic cancer patients compared with the corresponding matching normal tissue, and this result is consistent with previous findings. However, the regulation of inflammation in pancreatic cancer cells is still unknown.

At present, a variety of natural compounds reduce the development of pancreatic cancer and inflammatory response. Kim and co-authors found that kaempferol ameliorates pancreatitis [11], and Zhang et al. reported that it regulates the PI3K/AKT signaling pathway, consequently inhibiting the proliferation of pancreatic tumor cells [12]. In addition, Yu et al. reported that the flavone quercetin inhibited IL6-induced PAAD proliferation [13]. Additionally, curcumin inhibits pancreatic cancer cell invasion through IL-6/ERK/NF-κB axis [14]. Although curcumin has excellent anti-inflammatory and antioxidant pharmacological effects, its low water solubility, low stability, and low bioavailability reduces the possibilities to use it as anticancer drug. Therefore, in recent years, researchers have modified the structure of curcumin to obtain analogues that overcome these drawbacks. Our team dedicated its efforts to the development of anti-inflammatory drugs and, based on our previous work, a series of chemical modifications on curcumin have been performed, obtaining several curcumin analogues with excellent anti-inflammatory activity. C66 used in this work is a representative compound among them [9,15,16,17]. The ability of C66 to potentially abolish the inflammatory TME in pancreatic cancer was evaluated in this work, and the results revealed that the treatment with C66 significantly reduced the inflammatory factors. This reduction effectively reduced the inflammatory TME and inhibited the proliferation of pancreatic cancer cells, as well as their ability to migrate. These results support the theory that inhibition of inflammation might control tumor growth. However, these results were not yet clarifying how C66 mediated the inflammatory response of pancreatic cancer.

Notably, one of our previous studies pointed out that C66 inhibits high glucose-induced inflammatory response in diabetic cardiomyopathy through targeting JNK [9]. JNK is one of the key members in the MAPK signaling pathway. Previous study showed that JNK is highly activated in pancreatic cancer [7]. The comparison of JNK phosphorylation in the three pancreatic cancer cells selected in this experiment revealed that p-JNK levels were extremely high in PANC-1 and SW1990 cells. In contrast, p-JNK was at a low level in normal pancreatic cells as well as in BxPC3. Thus, JNK phosphorylation after the treatment with C66 in PANC-1 and SW1990 was evaluated, revealing that C66 inhibited JNK activation in a dose-dependent manner, and this dose-dependent result was consistent with the previous results on the inhibition of inflammatory factors. The subsequent experiments confirmed that the inhibition of JNK activation inhibited inflammatory TME and tumor proliferation and migration after JNK knockdown. This suggested that C66 could decrease the inflammatory TME in PAAD by regulating JNK, thereby inhibiting the proliferation and migration of pancreatic cancer cells.

A limitation of this study is that it does not further explore the potential reasons for the lower activation of JNK in the BxPC3 cell line compared with PANC-1 and SW1990. One possible hypothesis could be that BxPC3 is a KRAS wild-type cell line, whereas both PANC-1 and SW1990 are KRAS mutant cell lines. This difference suggests that the KRAS mutation could be associated with the level of JNK activation; although, this hypothesis needs to be confirmed.

## 4. Materials and Methods

### 4.1. Reagents and Chemicals

Compound C66 was obtained from our laboratory, and it was dissolved in DMSO. p-JNK, JNK, IL-6, and GAPDH antibodies were purchased from Proteintech (Wuhan, China). Horseradish peroxidase-conjugated anti-rabbit secondary antibodies were purchased from Abcam.

### 4.2. Cell Culture

Pancreatic normal cell line (HPNE) and pancreatic cancer cell lines (BxPC3, SW1990, and PANC-1) were purchased from the Shanghai Institute of Biochemistry and Cell Biology (Shanghai, China). HPNE, SW1990, and PANC-1 cell lines were cultured in DMEM (Thermo Fisher Biotechnology, Shanghai, China,) containing sodium bicarbonate (1.5 g/L) and glucose (4.5 g/L) and supplemented with 10% FBS (Thermo Fisher Biotechnology, Shanghai, China), 100 U/mL penicillin, and 100 mg/mL streptomycin (Thermo Fisher Biotechnology, Shanghai, China). BxPC-3 cell line was cultured in RPMI 1640 medium (Thermo Fisher Biotechnology, Shanghai, China) containing the same supplements as above. All cells were incubated in a humidified environment at 37 °C with 5% CO_2_.

### 4.3. JNK Knockdown

The oligonucleotides used for JNK knockdown were synthesized by Sangon Biotech (Shanghai, China) and then the shRNA sequences were constructed into the pLKO.1-puro. The targeting sequences JNK1-shRNA-I, JNK1-shRNA-II, JNK2-shRNA-I, and JNK2-shRNA-II were the following: GCCCAGTAATATAGTAGTAAA, GAGTCGGTTAGTCATTGATAG, CTAACTTATGTCAGGTTATTC, and AGGGATTGTTTGTGCTGCATT, respectively. The knockdown efficiency was determined by qPCR and Western blot. The shJNK-A consists of JNK1-shRNA-II and JNK2-shRNA-I, and the shJNK-B consists of JNK1-shRNA-II and JNK2-shRNA-II. The shJNK-C consists of JNK1-shRNA-I and JNK2-shRNA-I, and the shJNK-D consists of JNK1-shRNA-I and JNK2-shRNA-II. Then, lentiviruses were produced in HEK 293T cells after the transfection of shRNA, psPAX2, and pCMV-VSV-G by lipofectamine 3000. Subsequently, stable PANC-1- or SW1990-JNK knockdown cell lines were generated via lentiviral transduction, followed by puromycin selection (2 µg/mL).

### 4.4. Immunohistochemistry

Human PAAD tissues were obtained from the First Affiliated Hospital, Wenzhou Medical University (Wenzhou, China), and the ethics committee of the same hospital approved our protocol with human samples. Briefly, two PAAD samples and the matching adjacent normal tissues were collected. Pancreatic sections (5-μm thick) were deparaffinized, rehydrated, and treated with 3% H_2_O_2_ for 10 min at room temperature. The sections were washed thrice with PBST and blocked with 1% BSA in PBST for 30 min at room temperature. Next, the primary antibody against IL-6 (1:200, Affinity Biosciences, Liyang, China) was added to the pancreatic sections and incubated at 4 °C overnight. Next, the sections were cultured with HRP-conjugated secondary antibody (1:500, Cell Signaling Technology, Shanghai, China) for 1 h at room temperature. The color development was obtained using DAB, the nuclei were stained using DAPI, and the sections were counterstained using hematoxylin for 1 min and then covered with neutral resins. All sections were examined under an optical microscope (200× magnification; Nikon, Tokyo, Japan).

### 4.5. Western Blot Assay

Cells were lysed with RIPA buffer on ice, centrifuged at 12,000 g and 4 °C for 15 min, and the supernatant was collected. Total protein concentration was determined using the BCA assay kit following the manufacturer’s instructions (Thermo Fisher Biotechnology, Shanghai, China). Protein samples (20–30 μg) were separated by 10% SDS-PAGE gel electrophoresis and electro-transferred to a 0.45 μm polyvinylidene difluoride membrane (PVDF). Subsequently, the membrane was blocked in 1% TBST with 5% non-fatty milk for 1 h at room temperature, followed by the overnight incubation at 4 °C with the specific antibodies. The secondary antibody horseradish peroxidase-conjugated goat anti-rabbit or anti-mouse IgG (1:10,000, Abcam, Shanghai, China) was added to the membrane and incubated at room temperature for 1 h. The bands were visualized by the ECL detection kit (Thermo Fisher Biotechnology, Shanghai, China) and analyzed by Image J software.

### 4.6. Real-Time Quantitative PCR (qPCR)

PAAD cells were treated with C66 at different concentrations and incubated for 24 h. Total RNA was isolated from cells using the RNAiso reagent (Takara Biomedical Technology, Beijing, China). Then, the RNA was subjected to reverse transcription using a 1st Strand cDNA Synthesis SuperMix (Yeasen Biotechnology, Shanghai, China) and qPCR was performed using Universal Blue qPCR SYBR Green Master Mix (Yeasen Biotechnology, Shanghai, China). The primers of the following genes such as JNK1, JNK2, IL-1β, IL-6, IL-8, IL-12β, TNF-ɑ, COX-2, and β-actin were synthesized by Sangon Biotech (Shanghai, China) and their sequences are listed in Table 1. The relative gene expression of each gene was calculated using the 2-ΔΔCT method and normalized to the amount of β-actin.

### 4.7. Cell Proliferation

#### 4.7.1. CCK-8 Assay

The cells were seeded in 96-well plates at a density of 5000 cells/well and incubated overnight. The viability of the transfected pancreatic cells was determined using the Cell Counting Kit-8 assay (CCK-8; APExBIO Technology, Shanghai, China) at 24 and 72 h. A measure of 10 μL CCK-8 reagent was added to each well and incubated at 37 °C for 1 h. The absorbance was read at 450 nm using a microplate reader.

#### 4.7.2. Colony Formation Assay

A total of 500 cells in their log phase were seeded in 6-well plates, cultured in a complete growth medium, and incubated for 7–10 days at 37 °C under 5% CO_2_ environment. The colonies were fixed with 4% paraformaldehyde for 15 min and stained with 0.05% crystal violet for 1 h. The number of colonies was counted under a microscope and expressed as percentage.

### 4.8. Migration Assay

#### 4.8.1. Scratch Wound Assay

A total of 1 × 10^6^ pancreatic cancer cells were resuspended in culture medium with 10% FBS and seeded into 6-well plates. The scratches were created using a 200 µL pipette tip when the cells reached 90% confluence. Then, the medium was removed, and the cells were washed using PBS to remove the non-attached cells, while the ones that remained attached were cultured in serum-free DMEM. The width of the scratch was measured and images were captured after 24, 48, and 72 h.

#### 4.8.2. Transwell Assay

The pancreatic cells were resuspended in serum-free medium. A total of 1 × 10^5^ cells/well were added to the upper chamber of the transwell with or without Matrigel (BD Biosciences, San Jose, CA, USA) to determine the invasion or migration ability. The complete cell culture medium containing fetal bovine serum was added to the lower chamber and the cells were incubated for 24 h. The cells in the lower compartment were stained with crystal violet, 5 random fields were randomly selected, cells were counted, and the number of invading or migrating cells was calculated under light microscope.

### 4.9. Enzyme-Linked Immunosorbent Assay (ELISA)

PAAD cells (1 × 10^5^) were treated with C66 at different concentrations. After incubated for 24 h, the cell medium was collected and determined with the IL-6 ELISA kit, according to the manufacturer’s instructions (Bioscience, San Diego, CA, USA).

### 4.10. Statistical Analysis

Statistical analysis was performed using GraphPad Prism 9.0. The data were obtained from three independent experiments in every experimental group in the in vitro studies. The data obtained in each experiment were subjected to a normality test and subjected to statistical analysis conformed to normal distribution. The analysis of variance was conducted to compare different groups. Results were expressed as mean  ±  standard error of mean (SEM). A value of *p*  <  0.05 was considered statistically significant.

## 5. Conclusions

Overall, this work reported a high level of inflammation in the PAAD environment, and JNK had a strong correlation with its regulation. In addition, C66 inhibited the occurrence and development of pancreatic cancer through JNK that affected inflammatory TME. The inhibition of inflammatory signals in the TME could be one of the options in the treatment of pancreatic cancer in the future.

## Figures and Tables

**Figure 1 molecules-27-03076-f001:**
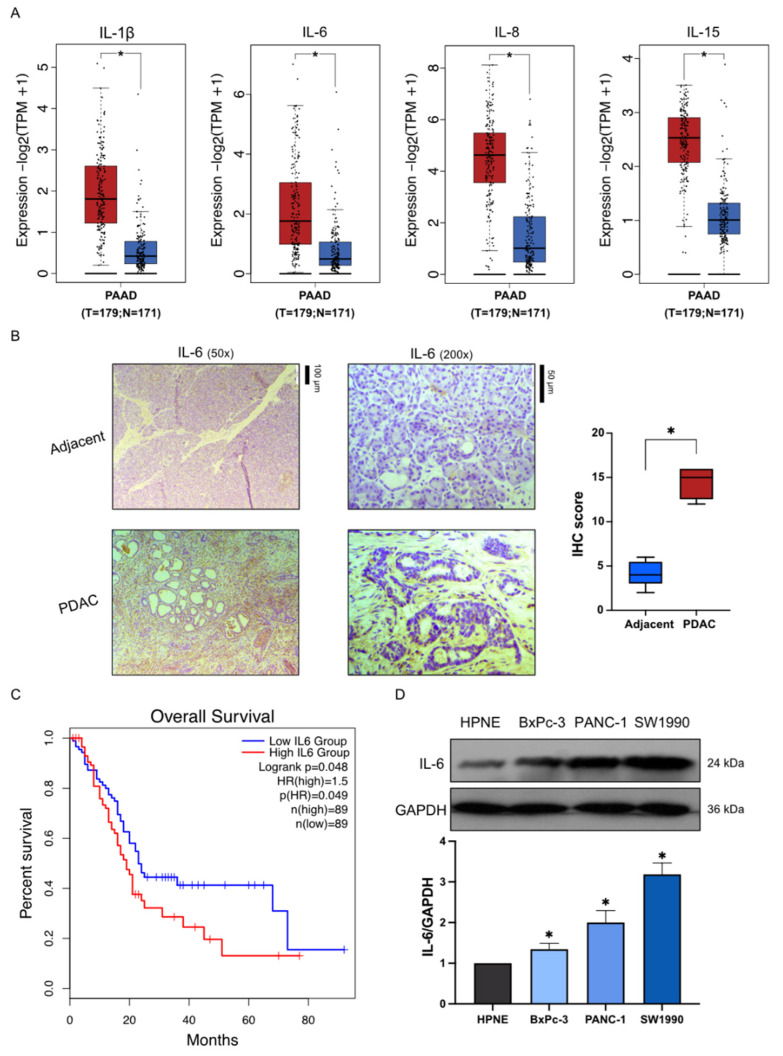
High-expression of inflammatory factors in pancreatic cancer. (**A**) The expression of IL-1β, IL-6, IL-8, and IL-15 in samples from pancreatic cancer patients (*n* = 179) and matching normal tissues (*n* = 171). (**B**) Immunohistochemical analysis of IL-6 expression in pancreatic cancer tissue and tumor-adjacent tissue. (**C**) The overall survival (OS) curves comparing patients with high and low IL-6 expression in PAAD. (**D**) Western Blot analysis of IL-6 expression in HPNE, BxPC-3, PANC-1, and SW1990. Each bar represents mean ± SEM of three independent experiments. *n* = 3, * *p* < 0.05, compared with the matching control.

**Figure 2 molecules-27-03076-f002:**
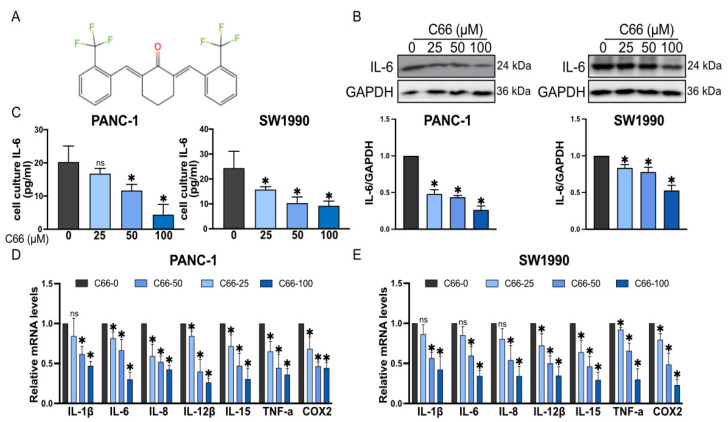
Inhibition of inflammatory factors by C66. (**A**) Structure of C66. (**B**) PANC-1 and SW1990 were treatment of with C66 (25, 50, and 100 μM) for 24 h. Protein levels of IL-6 were detected by Western blot. GAPDH was used as control. Densitometric quantification is shown in lower panel. Each bar represents mean ± SEM of three independent experiments. *n* = 3, * *p* < 0.05, compared with the matching control. (**C**) Levels of IL-6 in PANC-1 and SW1990 conditioned culture media as determined by ELISA. Cells were treated with C66 and IL-6 levels in media were measured at the indicated dose for 24 h. Each bar represents mean ± SEM of three independent experiments. *n* = 3, * *p* < 0.05, compared with the matching control; ns = not significantly different, compared with the matching control. (**D**,**E**) IL-1β, IL-6, IL-8, IL-15, TNF-α, and COX2 mRNA expression after the treatment of PANC-1 and SW1990 with C66 (25, 50, 100 μM) by RT-qPCR. Each bar represents mean ± SEM of three independent experiments. *n* = 3, * *p* < 0.05, compared with the matching control; ns = not significantly different, compared with the matching control.

**Figure 3 molecules-27-03076-f003:**
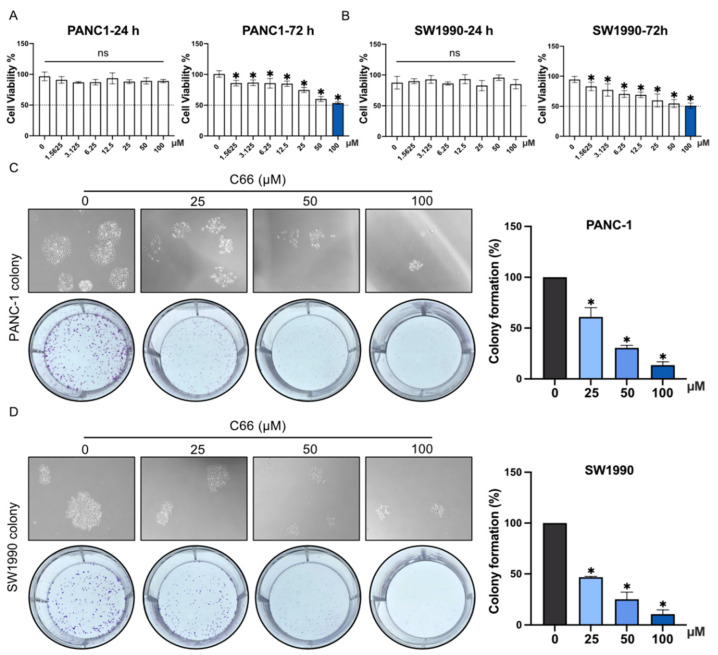
Inhibition of pancreatic cancer cell proliferation by C66. (**A**,**B**) PANC-1 and SW1990 proliferation was determined by CCK-8 assay after the treatment with C66 (0, 1.5625, 3.125, 6.25, 12.5, 25, 50, or 100 μM) for 24 or 72 h. Each bar represents mean ± SEM of three independent experiments. *n* = 3, * *p* < 0.05, compared with the matching control; ns = not significantly different, compared with the matching control. (**C**,**D**) PANC-1 and SW1990 colony formation after the treatment with C66 (25, 50, or 100 μM) administered for 7 days. Each bar represents mean ± SEM of three independent experiments. *n* = 3, * *p* < 0.05, compared with the matching control.

**Figure 4 molecules-27-03076-f004:**
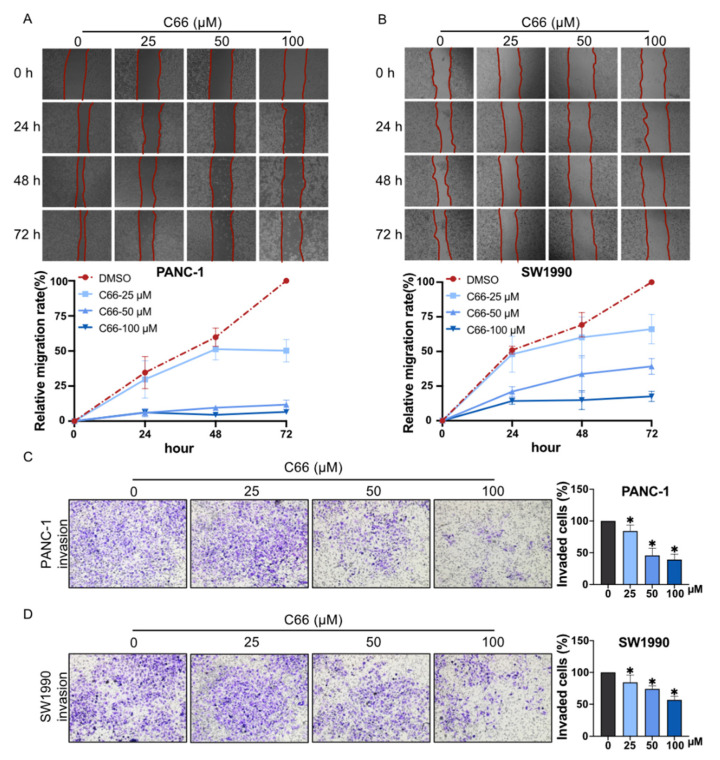
Inhibition of the migration of pancreatic cancer cells by C66. Scratch wound (**A**,**B**) and transwell (**C**,**D**) assay assessed the inhibition of cell migration after C66 (25, 50, or 100 μM) treatment of PANC-1 and SW1990 for 3 days. Each bar represents mean ± SEM of three independent experiments. *n* = 3, * *p* < 0.05, compared with the matching control.

**Figure 5 molecules-27-03076-f005:**
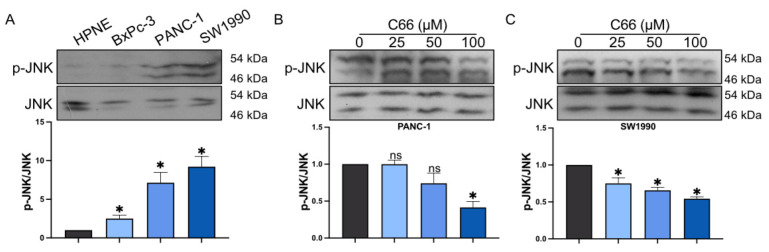
Reduction in pancreatic cancer cell JNK activation by C66 targeting. (**A**) JNK activation in HPNE, BxPC-3, PANC-1, and SW1990. (**B**,**C**) JNK activation measured in PANC-1 and SW1990 cells after the exposure to different doses of C66 (25, 50, or 100 μM) for 12 h. Each bar represents mean ± SEM of three independent experiments. *n* = 3, * *p* < 0.05, compared with the matching control; ns = not significantly different, compared with the matching control.

**Figure 6 molecules-27-03076-f006:**
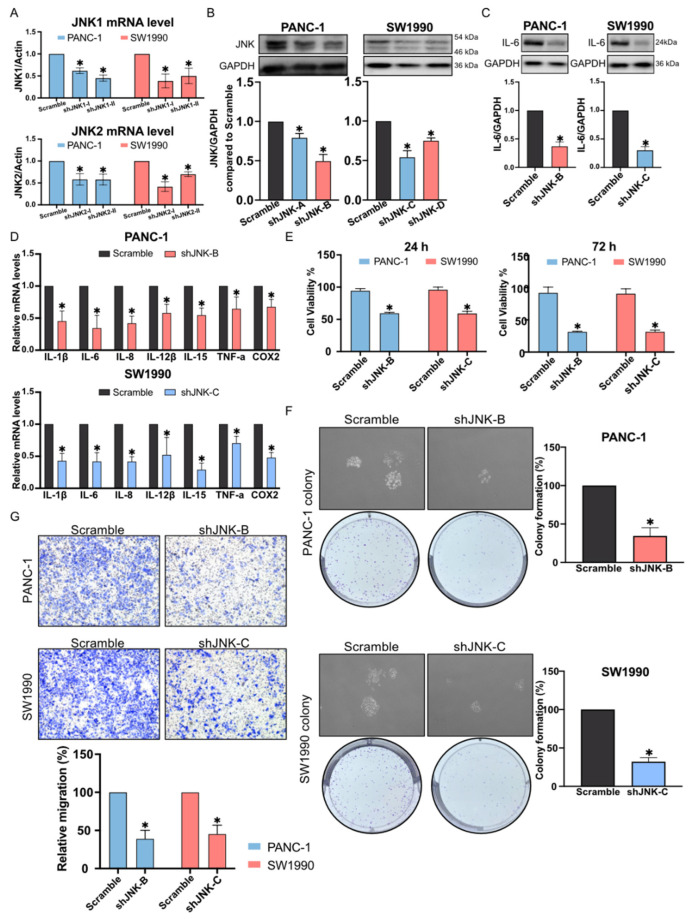
JNK knockdown inhibited cell inflammation and proliferation of pancreatic cancer cells. (**A**,**B**) JNK knockdown efficiency by RT-qPCR and Western blot after transfection of PANC-1 and SW1990 with shJNK-A, -B, -C, and -D. Each bar represents mean ± SEM of three independent experiments. *n* = 3, * *p* < 0.05, compared with the matching control. (**C**) IL-6 expression by Western blot after JNK knockdown in PANC-1 and SW1990. Each bar represents mean ± SEM of three independent experiments. *n* = 3, * *p* < 0.05, compared with the matching control. (**D**) IL-1β, IL-6, IL-8, IL-12β, IL-15, TNF-α, and COX2 mRNA expression by RT-qPCR after JNK knockdown in PANC-1 and SW1990. Each bar represents mean ± SEM of three independent experiments. *n* = 3, * *p* < 0.05, compared with the matching control. (**E**) PANC-1 and SW1990 proliferation was determined by CCK-8 assay after the JNK knockdown. Each bar represents mean ± SEM of three independent experiments. *n* = 3, * *p* < 0.05, compared with the matching control. (**F**) PANC-1 and SW1990 colony formation after the JNK knockdown for 7 days. Each bar represents mean ± SEM of three independent experiments. *n* = 3, * *p* < 0.05, compared with the matching control. (**G**) The transwell assay assessed the inhibition of cell migration after the JNK knockdown for 3 days. Each bar represents mean ± SEM of three independent experiments. *n* = 3, * *p* < 0.05, compared with the matching control.

**Table 1 molecules-27-03076-t001:** Primer sequences for qPCR.

Gene	Species	Sequence
JNK1	Human	ACACCACAGAAATCCCTAGAAG CACAGCATCTGATAGAGAAGGT
JNK2	Human	ATCAGAATCTGAGCGAGACAAA CAAACAGTGATGTATGGGTGAC
IL-1β	Human	GCCAGTGAAATGATGGCTTATT AGGAGCACTTCATCTGTTTAGG
IL-6	Human	CACTGGTCTTTTGGAGTTTGAG GGACTTTTGTACTCATCTGCAC
IL-8	Human	AACTGAGAGTGATTGAGAGTGG ATGAATTCTCAGCCCTCTTCAA
IL-12β	Human	TAAGATGCGAGGCCAAGAATTA TACTCATACTCCTTGTTGTCCC
TNF-a	Human	TGCACTTTGGAGTGATCGGC ACTCGGGGTTCGAGAAGATG
COX2	Human	TGTCAAAACCGAGGTGTATGTA AACGTTCCAAAATCCCTTGAAG
β-actin	Human	GATTCCTATGTGGGCGACGA AGGTCTCAAACATGATCTGGGT

## Data Availability

All data supporting the findings in our study are available from the corresponding author upon reasonable request.

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
