# Peer review of "Curcumin Derivative C66 Suppresses Pancreatic Cancer Progression through the Inhibition of JNK-Mediated Inflammation"

_molecules, 2022, doi:10.3390/molecules27103076_

Round 1
Reviewer 1 Report
Jiang et al., present a nicely designed study to evaluate the curcumin-derived compound C66 on pancreatic cancer. The authors show that a pro-inflammatory environment is present in PAAD and that PAAD cells contain elevated levels of activated JNK as well as produce elevated levels of inflammatory cytokines when compared to controls. Treatment with C66 as well caused a reduction in p-JNK, while treatment with C66 or knockdown of JNK resulted in decrease expression of inflammatory cytokines by the PAAD cell lines. The study is nice and rather complete. There are a few concerns to be addressed.
- In section 2.1 of the Results, the authors should give a more thorough explanation leading up to the experiment presented in Figure 1 (i.e., What tissues were analyzed? How many samples/patients were analyzed? What was the tumor grading of the samples? Were the normal (control) tissues isolated from the same patient (matched normal)? Some of this information is present in the figure, but it should also be included in the text.
- Please state the number of times the experiment in Figure 1D was repeated in the Figure legend.
- In the legend of Figure 2, please state for how long the cells were treated with C66. How many experimental replicates are represented?
- The IC50 for C66 is relatively high. What is known about the bioavailability and stability of C66? Can it reach therapeutic levels in vivo, in its current form?
- In the legend of Figures 3, 4, 5 and 6 please state the number of replicates represented.
- In the legend of Figure 5, please state for how long the cells were treated with C66.
- In section 4.4, the number of samples/patients analyzed should be stated.
- Line 286 refers to Table 1 for primers. There is no Table 1 presented in the manuscript.
- There are some minor typos:
- Line 47, "in 2030" to "by 2030".
- Line 70-71, should read, "JNK, a member of the MAPK family, is in an abnormally activated state in PAAD"
- Line 94, "expression in the pancreatic..."
- Line 143, "was increased as compared to the control cell line, HPNE"
- Line 212, "results"
- The numbers of scientific notation on lines 305 and 312 should be superscripted.
- Line 328, "inflammation in the PAAD environment"
Reviewer 2 Report
The authors analyzed the role of C66 in the proliferation and migration of pancreatic cancer. The authors demonstrated decrease in IL6 protein expression after CD66 therapy, reduction of migration after CD66 therapy and reduction in JNK activation. The manuscript is well-written, and data is clearly presented.
Comments to the authors:
- Please put into figure 1A legend how many normal (n=171) and tumor tissue samples (n=179) were analyzed. It’s hard to read sample size on the chart.
- Figure 1B. Photograph magnification is too high to see any staining detail. Please provide higher magnification of IL6 staining for tumor tissue and adjacent normal tissue
- The authors do not evaluate extracellular IL6 levels after CD66 or JNK knockdown. Figure 2B. Data demonstrating IL6 drop in extracellular space after CD66 treatment would enhance authors conclusions.
- My main concern is with the statement in the abstract that “The results demonstrated that JNK knockdown could significantly inhibit the proliferation and migration of pancreatic cancer” The authors did not show either effect on proliferation or migration after JNK knockdown in this manuscript. There is an effect of CD66 on migration and proliferation and also JNK, but as any chemical compound used at high concentration it can have effect on multiple proteins and JNK is only one of them. The authors need to provide the data to support the statement.
Round 2
Reviewer 2 Report
The authors addressed all questions in depth.